# Stress-Insensitive Resonant Graphene Mass Sensing via Frequency Ratio

**DOI:** 10.3390/s19133027

**Published:** 2019-07-09

**Authors:** Xing Xiao, Shang-Chun Fan, Cheng Li, Wei-Wei Xing

**Affiliations:** 1School of Instrumentation and Optoelectronic Engineering, Beihang University, Beijing 100191, China; 2Key Laboratory of Quantum Sensing Technology (Beihang University), Ministry of Industry and Information Technology, Beijing 100191, China

**Keywords:** stretched graphene resonator, mass sensor, resonant mode, frequency ratio, molecular dynamics simulation

## Abstract

Herein, a peripherally clamped stretched square monolayer graphene sheet with a side length of 10 nm was demonstrated as a resonator for atomic-scale mass sensing via molecular dynamics (MD) simulation. Then, a novel method of mass determination using the first three resonant modes (mode11, mode21 and mode22) was developed to avoid the disturbance of stress fluctuation in graphene. MD simulation results indicate that improving the prestress in stretched graphene increases the sensitivity significantly. Unfortunately, it is difficult to determine the mass accurately by the stress-reliant fundamental frequency shift. However, the absorbed mass in the middle of graphene sheets decreases the resonant frequency of mode11 dramatically while having negligible effect on that of mode21 and mode22, which implies that the latter two frequency modes are appropriate for compensating the stress-induced frequency shift of mode11. Hence, the absorbed mass, with a resolution of 3.3 × 10^−22^ g, is found using the frequency ratio of mode11 to mode21 or mode22, despite the unstable prestress ranging from 32 GPa to 47 GPa. This stress insensitivity contributes to the applicability of the graphene-based resonant mass sensor in real applications.

## 1. Introduction

As typical representatives in the burgeoning field of nanoelectromechanical systems (NEMS), nanomechanical resonators [1,2,3] are expected to bring dramatical improvement to mass [4,5,6,7] pressure [8], acceleration [9] measurement and chemical/biological detections [10,11,12] due to their ultra-high sensitivity. Significantly, the mass sensitivity reached 10^−22^ gHz^−1/2^ by using a 205 nm long carbon-nanotube (CNT) as a resonator [6]. In other words, the mass of a single gold atom can be measured. The measure principle of a nanomechanical resonant mass sensor is generally based on the frequency shift induced by particles absorbed in the resonant beam or a membrane made from carbon nanotubes (CNT) or graphene [4,5,6,7,13,14]. In addition to their excellent mechanical properties, such as extremely high intrinsic strength and Young’s modulus [15,16] and tunable electrical performance [17,18,19], graphene sheets have a larger aspect ratio compared with CNTs, which means that enough areas are available for incoming mass flux. Consequently, an increasing number of theoretical studies have focused on graphene-based nanomechanical resonators with atomic-mass resolution in recent years [4,20,21,22,23,24,25,26,27,28].

It should be noted that nonlocal elasticity theory [21,22], molecular structural mechanics methods [20], and molecular dynamics (MD) simulation methods [25,26,27,28] have been studied to calculate the frequency shift of graphene membrane in response to the absorbed mass. Among these methods, MD simulation shows an extraordinary advantage in investigating the nanostructure’s properties by directly computing the state of every atom, which takes the scale effect into consideration. In fact, using MD simulation, Arash et al. [25] studied the frequency shifts of graphene sheets with four edges fixed and two edges fixed due to attached gas atoms, and the mass sensitivity of a square single-layer graphene sheet with a side length of 10 nm was achieved to reach 10^−21^ gHz^−1/2^. Then Kwon et al. [26] explained the gate voltage’s effect on the mass induced frequency shift, which was in good agreement with the experimental results of Chen et al. [4]. Recently, Duan et al. [27] proposed a new kind of resonator with a mass resolution of 10^−24^ g using pillared graphene structures. However, these studies focused on either relaxed graphene sheets or graphene sheets with tension induced by the electrostatic force between graphene and silicon oxide substrate, which was far below the intrinsic strength (about 130 GPa) [16]. In fact, stretched graphene sheets with greater prestress, tended to have higher resonant frequencies, and were expected to achieve higher mass sensitivity. Moreover, in previous studies, the calculation of the absorbed mass was mainly based on fundamental frequency shift, which is sensitive to stress in graphene. Nevertheless, it is not easy to control the stress in stretched graphene accurately and steadily. For example, the temperature fluctuation, resulting from the difference of the thermal expansion coefficient between graphene and silicon oxide substrate, disturbs the stress in graphene significantly [1]. In addition, the variance of the gate voltage may influence the stress as well [4]. 

However, it is important to mention that higher resonant modes of graphene, which are common in the vibration process [29,30], can be used to compensate for the fundamental frequency shift induced by unstable stress. Therefore, in this paper, the applicability of a stretched graphene-based mass sensor via the frequency ratio of the first three resonant modes was investigated by performing MD simulation. A peripherally clamped square graphene sheet with a side length of 10 nm served as the resonator, and the frequencies, as well as the mass-induced frequency shifts of the three resonant modes (mode11, mode21, and mode22), were calculated. Considering the susceptibility of fundamental frequency to unstable stress in a stretched graphene sheet, a novel method of mass determination based on the frequency ratio of mode11 to mode21 or mode22 was proposed, where the frequencies of mode21 and mode22 were used to compensate for the fundamental frequency shift caused by stress variation.

## 2. Modeling and MD Simulation

As shown in Figure 1a, the resonant mass sensor is mainly composed of the substrate and the square graphene sheet. The substrate is a silicon wafer with a SiO_2_ layer on the top. A trench is etched into the central SiO_2_. Then a gate electrode, a source electrode and a drain electrode are placed at the bottom and on the two sides of the trench. Thus, the gate electrode sends an actuation force to the graphene sheet, while the source and the drain electrodes are used to detect vibration [1,4,29,30,31,32]. In this case, the graphene sheet is deposited on the trench. Note that the adhesion between the graphene and the substrate is strong enough (up to 20.64 J/m^2^) to clamp the four edges of the graphene [33,34]. Moreover, when strong tension was exerted to the graphene in this study, as shown in Figure 1b, enlargement of the area of graphene fixed prevents slippage between the graphene and the silicon oxide substrate. When external pressure is applied to the silicon wafer, the deformation of the silicon oxide substrate stretched the graphene, therefore adjusting the tensile stress in the graphene. 

To investigate the resonant frequency of the stretched graphene, the evolutionary process of vibrating the graphene was obtained by calculating every atom’s position and momentum directly [35], which was performed with the Large-scale Atomic/Molecular Massively Parallel Simulator (LAMMPS) package [36]. As shown in Figure 1b, the square monolayer graphene sheet contained 7782 atoms with 3928 fixed atoms (boundary) and 3854 free atoms (middle, 10 nm × 10 nm). The four edges were fixed herein, thus preventing the “flipping” motion, which generally happens with free edges, and decreasing the resonator’s quality factor significantly [30,37,38,39]. Then, gold atoms were added to this system as absorbed mass. The adaptive intermolecular reactive empirical bond order (AIREBO) [40,41] and the potential and embedded-atom method (EAM) [42] potential were used to describe C–C and Au–Au interactions, respectively. The Lennard–Jones 12–6 equation, representing a cursory approximation of the interaction between C and Au [28,43,44,45], is defined by
(1)$$E_{L-J}=4\epsilon \left[{\left(\frac{\sigma }{r}\right)}^{12}-{\left(\frac{\sigma }{r}\right)}^{6}\right]$$
where *ϵ* = 29 meV and *σ* = 3.0 Å. With these three potential functions, the Hamiltonian equation of the system could be established. By solving the Hamiltonian equation, the corresponding evolutionary process could be determined after obtaining every atom’s position and momentum.

The simulation process was divided into four parts: equilibration, deformation, actuation and oscillation. Followed by further equilibration under the NPT ensemble (where the number of atoms, pressure and temperature are kept constant) of 100 ps, the simulation system was first optimized to obtain the relaxed structure with the minimum energy. Then, the axial deformation was imposed on the graphene sheet to stretch it. The induced stress ranged from 12 GPa to 47 GPa in this study. Afterwards, the edges were fixed, and then a velocity distribution *v_z_* was exerted on the free part, described as
(2)$$v_{z}=v_{0}\sin\left(n\frac{x}{L_{x}}\pi \right)\sin\left(m\frac{y}{L_{y}}\pi \right)$$
where *m* and *n* refer to the resonant mode, *x* and *y* are the coordinates of every atom on the *x–y* plane, *L_x_* and *L_y_* are the side length of the vibrating square graphene sheet, which were both set to 10 nm. The initial velocity *v*_0_ is equal to 1 Å/ps. In this case, the corresponding amplitude was less than 1.5 Å, which is too small to cause violent nonlinear vibrations [38,44]. We made a convergence test, which demonstrates that 1 Å/ps is appropriate for harmonic vibration, and the corresponding results are shown in Section 3.1. After the graphene sheet started oscillating under the NVE ensemble (where the number of atoms, volume and energy are kept constant), the kinetic energy and potential energy of the system were traced and recorded. Finally, the Fast Fourier Transform (FFT) method was utilized to calculate the frequency of the energy change, which was twice the vibration frequency. The whole MD simulation lasted for 600 ps with a time-step of 1 fs, and the simulation temperature was 10 K.

## 3. Results and Discussion

By performing the MD simulation, we calculated three corresponding resonant frequencies of the peripherally clamped graphene sheet with different absorbed masses and prestress values. The simulation results show that the position of the absorbed mass greatly affects the frequency shift. Particularly, when gold atoms were placed at the center area of the graphene sheet, the frequency of mode11 decreased significantly, whilst the frequencies of mode21 and mode22 remained steady. Moreover, a linear relationship exists between the frequency and the square root of tension in a graphene sheet, thus indicating an effective way to improve the mass-induced frequency shifts by increasing the prestress. However, this phenomenon also means that stress fluctuation would disturb the determination of the absorbed mass to a large extent. Consequently, a novel and effective method for solving the absorbed mass is proposed based on the ratio between the fundamental frequency *f*_11_ and the higher frequencies *f*_2*1*_, and *f*_22_, instead of the fundamental frequency shift susceptible to stress variance. 

### 3.1. Effect of Absorbed Mass Distribution

Since the bending rigidity of the monolayer graphene is as low as 2.31 × 10^−19^ Nm [46], the stretched monolayer graphene sheet, as shown in Figure 1b, can be modeled as a flat square membrane under tension [30], whose resonant frequencies can be written as
(3)$$f_{nm}=\frac{\sqrt{n^{2}+m^{2}}}{2L}\sqrt{\frac{\sigma }{\rho_{0}}}$$
where *L* is the side length of the square membrane, *σ* is the stress of graphene, and *ρ*_0_ is the density. The parameters *n* and *m* refer to the different resonant modes as defined in Figure 2. Considering that Equation (3) is applicable to harmonic vibration, a convergence test was made beforehand to ensure that the nonlinearity was neglectable in our simulation. A set of amplitudes, ranging from 0.2 to 10 Å/ps of initial velocity, were applied successively, and the corresponding frequencies of mode11, mode21 and mode22 were then obtained by the MD simulation, as shown in Figure 3. In this study, the stretched graphene sheets with a prestress ranging from 12 to 47 GPa serve as resonators. Since a strong prestress in the graphene can decrease the out-of-plane deflection dramatically and weaken the nonlinearity, the value of 12 GPa was chosen in this convergence test to render the simulation results effective. Figure 3 shows that when the initial velocity amplitude was lower than 2 Å/ps, the error between the MD simulation results and the theoretical results obtained in Equation (3), based on the harmonic vibration hypothesis, was within 5%. The small variance of resonant frequencies demonstrates the convergence of the MD simulation results, thus indicating that the nonlinearity effect was weak in this case. Consequently, the initial velocity amplitude was set to 1 Å/ps and the corresponding vibration can be regarded as harmonic.

When the gold atoms were put on the surface, the equivalent density of graphene changed slightly and the corresponding frequency shift can be described as
(4)$$\frac{\Delta f_{nm}}{f_{nm}}=-\frac{1}{2}\frac{\Delta m}{m_{0}}$$
The volume of the graphene sheet is considered unchanged, so the change of density can be regarded as the change of mass. Note that this equation is suitable for evenly distributed golden atoms. Otherwise, a correction factor *S_nm_* (*x*, *y*) should be added in Equation (4) [47]. The correctional formula is expressed as
(5)$$\frac{\Delta f_{nm}}{f_{nm}}=-\frac{1}{2}\frac{\Delta m}{m_{0}}S_{nm}(x,y)$$

The factor *S_nm_* (*x*, *y*) represents the effect of the absorbed mass distribution. In order to obtain the value of *S_nm_* (*x*, *y*) corresponding to each position, we divided the graphene sheet into 100 small square areas, and two gold atoms were placed on each of them in turn. The resulting frequency shift was then calculated by the MD simulation. After the traversing of all 100 areas, the results are illustrated in Figure 4. For this part, the prestress was set to 47 GPa, so the stress caused by oscillation (about 0.2 GPa) can be neglected. The frequencies of mode11, mode21 and mode22 were 323.2 GPa, 515.4 GPa, and 647.4 GPa respectively, which agrees well with the theoretical values of *f*_11_ = 326.8 GPa, *f*_21_ = 516.8 GPa, and *f*_22_ = 462.2 GPa. When two gold atoms were placed on the graphene surface, the corresponding frequency shifts Δ*f_nm_* were obtained using the MD simulation. As for the right side of Equation (5), the vibrating part of the graphene contained 3854 carbon atoms with a molar mass of 12, and the molar mass of the two absorbed gold atoms was 197, therefore calculating Δ*m/m*_0_ as 0.0085. In this way, the value of *S_nm_* (*x*, *y*) can be confirmed by Equation (5), as illustrated in Figure 4.

Referring to Figure 4a, the mass in the central area reduced the fundamental frequency dramatically with the correction factor *S*_11_(0, 0) of about 4, while the mass at the edge had little influence on the frequency. Note that the areas with larger vibration amplitudes tended to have a larger factor *S_nm_*(*x*, *y*), which was similar to the deflection eigenmode. Hence, *S_nm_*(*x*, *y*) could be written as *S_nm_*(*x*, *y*) = 4[1-cos(2*n*π*x*/*L_x_*)][1-cos(2*m*π*y*/*L_y_*)] [48]. This expression conforms well with the simulation results, which had a coefficient (*R*^2^) of determination of up to 0.98 for the mode11. This conclusion can also be applied to higher modes (mode21 and mode22), as shown in Figure 4b,c. However, the fitting results are not very accurate because they are only a coarse approximation of the deflection eigenmode of mode21 (*n* = 2, *m* = 1) and mode22 (*n* = 2, *m* = 2). Compared with the one in mode11, the areas with large *S_nm_* (*x*, *y*) values in mode21 and mode22 are much smaller and more scattered, which produces an adverse effect on the detection of the absorbed mass. Therefore, the frequency shifts of mode21 and mode22 are not available for direct mass determination. In contrast, the frequencies of these higher modes can be used to compensate for the effects of stress instability in mass determination using the fundamental frequency shift. It should be mentioned that the prestress of stretched graphene cannot remain entirely unchanged. Although the stress fluctuation affects the frequencies of all the modes, the absorbed mass in the center only reduces the fundamental frequency. As a result, the frequencies of mode11, mode21 and mode22 were employed together to determine the absorbed mass accurately in spite of the unstable stress in the graphene. 

### 3.2. Effect of Prestress Variation

It is well known that enhancing the stress in graphene sheets contributes to an increase of rensonant frequencies and frequency shifts. Hence, the square graphene sheet was stretched axially to generate a series of tensile stress at a range of 12–47 GPa. To achieve higher sensitivity, absorbed gold atoms were constrained to a circular area with a radius of 10 Å in the middle of the graphene sheet. Then, the resulting different frequency responses to the absorbed mass were observed, as depicted in Figure 5. Under tensile stresses of 12, 23, 32, 40 and 47 GPa, the fundamental natural frequency of the graphene sheet was calculated as 169.5, 227.3, 268.0, 298.8 and 323.2 GPa, respectively, as shown in Figure 5a. The fundamental frequency is proportional to the square root of the tensile stress, as indicated in Equation (3). Moreover, Figure 5a presents the frequency shift of mode11 when 1–10 gold atoms with masses of 3.3–33 × 10^−22^ g were placed in the middle of the monolayer of the graphene sheet. The fundamental frequency shift was −0.480, −0.643, −0.772, −0.864 and −0.943 GHz/(10^−^^22^ g), respectively, thus showing a linear relashionship with the square root of the tensile stress. By contrast, Figure 5b,c show the insensitivities of mode21 and mode22 to central mass. The frequency shifts of mode21 and mode22 were less than 0.01 and 0.05 GHz/(10^−22^ g), respectively, and much lower than that of mode11. It can also be seen from Figure 5b,c that the frequencies of mode21 and mode22 increased simultaneously with the increasing stress in the graphene, obeying Equation (3), as the fundamental frequency did. In Figure 5d, the frequency-shift ratio Δf_nm_/f_nm_ under different tensions exhibited almost an identical relationship with the absorbed mass. The average frequency shift ratio of mode11 induced by the central absorbed mass of 10^−22^ g is 0.287%, similar to the theoretical result of 0.266% and in accordance with Equation (5) where S_11_(0, 0), is assumed to be 4. On the contrary, the frequencies of mode21 and mode22 showed negligible variations in response to the absorbed mass. In this way, the insensitivity to the central absorbed mass can be made use of to compensate for the influence of stress fluctuation.

### 3.3. Mass Determination by Frequency Ratio

In previous studies [6,7,9,10,11,22,23,24], the determination of the absorbed mass was simply based on the fundamental frequency shift or the fundamental frequency shift ratio. For the use of the former, the change of the tension in the graphene would induce a significant interference with the calculation of the absorbed mass. For the use of the latter, the natural frequency *f*_0_ is generally obtained in advance, but other factors, such as temperature, are likely to change during the measurement of the resonant frequency *f* after mass absorption. The frequency shift ratio (*f* − *f*_0_)/*f*_0_ results from the absorbed mass and the circumstance change. There is no essential difference between these two methods. As a result, these methods of mass determination are inapplicable to the stretched graphene-based resonator, whose stress cannot be controlled precisely and steadily.

Herein, the absorbed mass can be determined using the frequency ratio of mode11 to mode21 or mode22, which are measured simultaneously, while suppressing the disturbance induced by the variance of tension. To achieve higher sensitivity, absorbed gold atoms were constrained to the middle of the graphene sheet, as demonstrated in Section 3.2. In this way, the parameter S_11_(0, 0) was up to 4, while S_21_(0, 0) and S_22_(0, 0) were equal to 0. From Equations (3) and (5), the absorbed mass can be calculated as
(6)$$\Delta m=\frac{m_{0}}{2}(1-\frac{\sqrt{10}}{2}\frac{f_{11}}{f_{21}}),$$
or
(7)$$\Delta m=\frac{m_{0}}{2}(1-2\frac{f_{11}}{f_{22}})$$
As shown in Equations (6) and (7), since the absorbed mass is linearly dependent on the frequency ratio of mode11 to mode21 (or mode22), a linear function is employed to fit the MD simulation results as given in Figure 6, which demonstrates a high correlation coefficient of 0.999. It should be noted that the parameters changed slightly when the prestress was not strong enough, especially when it was comparable to the effective stress due to the nonlinear oscillations (about 0.2 GPa). As a consequence, only the frequencies under stresses of 32, 40 and 47 GPa are depicted in Figure 6. Furthermore, theoretically, only the mass of the graphene sheet in Equations 6 and 7 were undetermined. In other words, the stretched graphene-based resonant mass sensor can reach an atomic scale resolution via the frequency ratio of mode11 to mode21 or mode22, which is advantageously insusceptible to stress fluctuation.

In addition, the absorption of the gold atoms on the surface of the graphene sheet needs to be mentioned. In this study, lateral movements of gold atoms were constrained, in accordance with the assumption that gold atoms are fixed tightly at the simulation temperature of 10K. However, the diffusion of gold atoms could not be neglected with the increase in temperature. The mass-sensing capability was inclined to worsen particularly when the temperature reached up to 300K due to the diffusion of gold atoms [45]. Moreover, increasing temperature leads to a decrease in the quality factor [1,4], which reduces the resolution of frequency shifts. Consequently, the conclusions obtained at low temperatures cannot be generalized to high temperatures without any modifications. Future research on restraining the diffusion and maintaining the mass sensing capability effectively under high temperature is needed. 

## 4. Conclusions

In this paper, the applicability of a stretched graphene-based mass sensor via a frequency ratio was investigated by performing an MD simulation. With regard to the square graphene sheet peripherally clamped, the frequencies and the mass-induced frequency shifts of mode11, mode21, mode22 were analyzed. The simulation results show that, on the one hand, absorbed mass in areas with a larger vibration amplitude decreased resonant frequencies more dramatically. The frequency shift of mode11, induced by the central absorbed mass was four times larger than that induced by evenly distributed mass, while the frequencies of mode21 and mode22 were totally insensitive to the central absorbed mass. On the other hand, a strong linear relationship between the frequencies and the square root of stress in graphene was found; thus, the stretched graphene sheet tended to have higher resonant frequencies and higher sensitivities. The fundamental frequency of a 10 nm long square monolayer graphene sheet with a prestress of about 47 GPa was up to 323.2 GHz, which exhibited an ultra-high mass sensitivity of 0.943 GHz/(10^−22^ g). Compared with the inapplicable traditional method of mass determination based on the fundamental frequency shift due to the unstable stress in stretched graphene, the proposed method of mass determination via the frequency ratio of mode11 to mode21 or mode22 can achieve a mass resolution of 3.30 × 10^−22^ g, with an unstable stress ranging from 32 GPa to 47 GPa. The resulting mass sensitivity was about 0.183 %/(10^−22^ g) for *f*_11_/*f*_21_ and 0.142 %/(10^−22^ g) for *f*_11_/*f*_22_. The benefit of stress immunity indicates the great robustness of the proposed sensor against external disturbances in real conditions. 

## Figures and Tables

**Figure 1 sensors-19-03027-f001:**
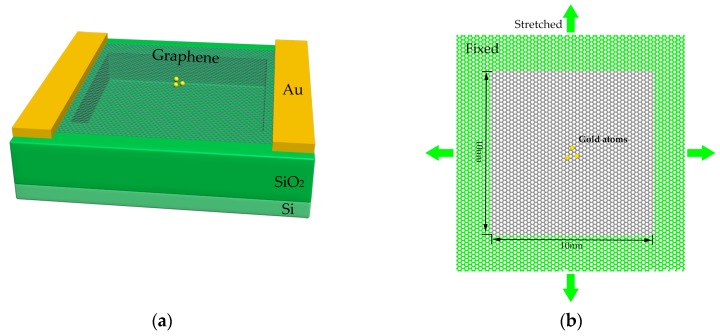
Schematics of the stretched graphene-based mass sensor. (**a**) A monolayer graphene sheet suspended on a silicon oxide substrate. (**b**) A peripherally fixed stretched monolayer graphene sheet with gold atoms absorbed on the surface.

**Figure 2 sensors-19-03027-f002:**
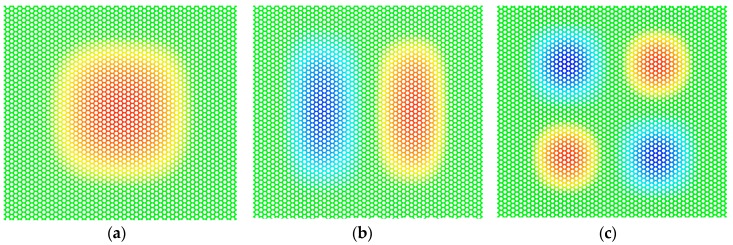
Three typical mode shapes of the square membrane peripherally clamped. (**a**) Mode11: n = 1, m = 1. (**b**) Mode21: n = 2, m = 1. (**c**) Mode22: n = 2, m = 2.

**Figure 3 sensors-19-03027-f003:**
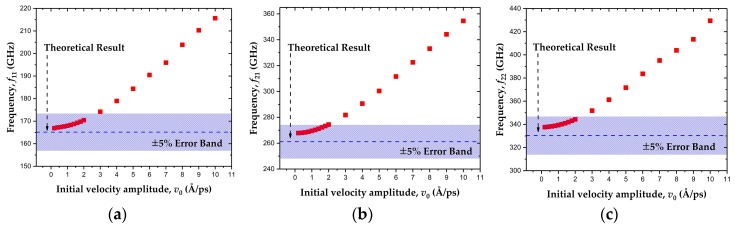
Frequencies calculated by the molecular dynamics (MD) simulation at different initial velocity amplitudes. (**a**) mode11, (**b**) mode21, (**c**) mode22.

**Figure 4 sensors-19-03027-f004:**
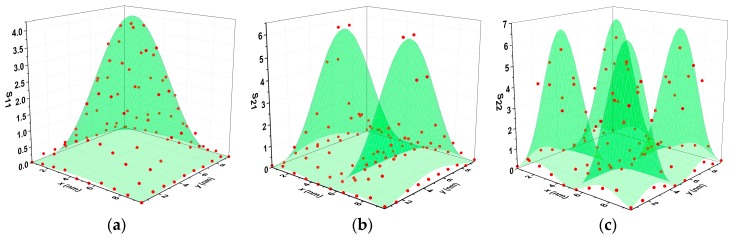
The values of *S_nm_* (*x*, *y*) obtained by MD simulation for (**a**) mode11, (**b**) mode21, and (**c**) mode22.

**Figure 5 sensors-19-03027-f005:**
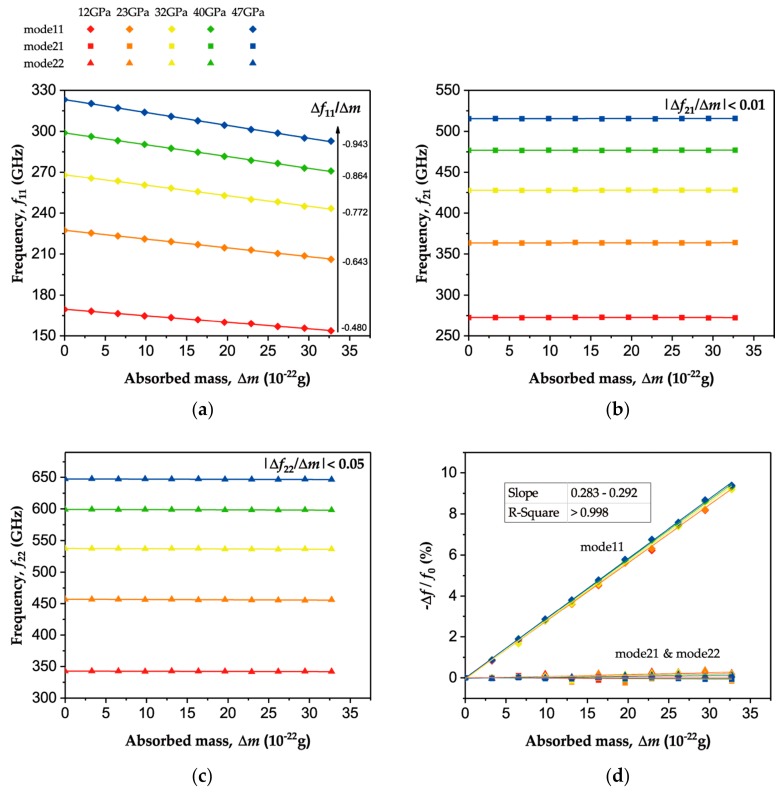
The frequency shift in response to the absorbed mass in the middle of the monolayer graphene sheet with a prestress of 12-47 GPa. The frequencies of (**a**) mode11, (**b**) mode21, and (**c**) mode22 versus absorbed mass. (**d**) The frequency shift rates Δ*f_nm_/f_nm_* versus absorbed mass.

**Figure 6 sensors-19-03027-f006:**
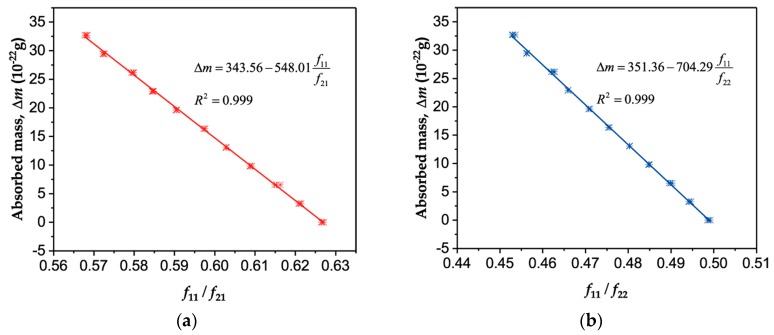
The relationship between the absorbed mass and the frequency ratio *f*_11_*/f*_21_, *f*_11_*/f*_22_. (**a**) The absorbed mass as a function of *f*_11_*/f*_21_. (**b**) The absorbed mass as a function of *f*_11_*/f*_22_.

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
