# Peer review of "Stress-Insensitive Resonant Graphene Mass Sensing via Frequency Ratio"

_sensors, 2019, doi:10.3390/s19133027_

Reviewer 1 Report

The manuscript authored by Xiao et al. presents the molecular dynamics simulation-based study of the sensing performance of graphene resonator, which operates in a few different vibration modes. Although the manuscript may be of interest to the community, the authors should consider following issues to revise the manuscript for further consideration:

1. The degree of vibration for a (monolayer) graphene resonator is determined by the magnitude of actuation force. In other words, when an actuation force is less than a certain critical value, the vibration mode of the graphene resonator becomes to correspond to the harmonic vibration. Meanwhile, for an actuation force being larger than a certain critical value, the vibration behavior of the graphene becomes the nonlinear vibration mode. In the paper, the authors only considered the harmonic vibration of the graphene resonator for studying its dynamics (i.e. frequency behavior) and mass-sensing performance. The author should specifically demonstrate how they can only consider the harmonic vibration behavior without taking into account the nonlinear effect in the vibration. In particular, the authors used the value of initial velocity applied to a graphene to induce the vibration of the graphene. What is the critical value of initial velocity in order to determine whether graphene resonator undergoes either harmonic or nonlinear vibration? Is there any particular reason to exclude the nonlinear vibration effect?

2. As the authors described that the location of atomic adsorption determines the frequency shift of a graphene resonator due to mass adsorption. I suspect that the authors assumed the atomic adsorption that occurs at the center of graphene. That is why the authors could observe only the frequency shift of graphene for mode11, while there is no frequency shift for mode21 and mode22. What happens if the atomic adsorption occurs at other locations? In other words, how about the frequency shift for each mode when atoms are adsorbed onto some location (rather than the center) of the graphene? In addition, the authors may be able to obtain the theoretical prediction of the frequency shift as a function of the location at which atomic adsorption occurs. For example, see Reference: Kim, et al., Beilstein J. Nanotechnol. 7, p.685-696 (2016); Dai, et al., Nanoscale Res. Lett. 7, Art. No. 499 (2012).

3. The authors provided the empirical equation for a relationship between the amount of adsorbed mass and the ratio of frequency mode11 to frequency mode21 (or mode22), i.e. Eq. (6) and (7). However, for a harmonic vibration, I think that the authors are able to obtain the theoretical equation for such a relationship. In other words, instead of using empirical parameters A, B, C, and D for Eq. (6) and (7), the authors could obtain such empirical parameters using the geometric parameters and physical properties of a monolayer graphene. I recommend the authors to present the theoretical equation for the relationship between the amount of adsorbed mass and the ratio of mode11 frequency to mode21 frequency (or mode22 frequency).

4. In the introduction, the author should cite the review papers, which discuss the current state-of-arts in nanomechanical resonators and their applications in mass sensing. Please cite the following review papers: Artlett et al., Nat. Nanotech. 6, p.203-215 (2011); Eom et al., Phys. Rep. 503, p.115-163 (2011); Waggoner and Craighead, Lab Chip 7, p.1238-1255 (2007)

Reviewer 2 Report

 The authors demonstrate a clamped stretched square monolayer graphene resonator for atomic scale mass sensing via molecular dynamics (MD) simulation. The authors demonstrate a very high sensitivity and free-stress measurements for mass added at the centre of the membrane, by measuring the mode 11 and the mode 21 and mode 22. It is a very interesting approach to increase the sensitivity.

The authors consider that the mass is added at the middle of the graphene layer, and only a mass change is produce. However, it has been previously demonstrated that the position of the added mass affect the resonance frequency due to the effect on the resonator stiffness.

I would recommend to include a bit of discussion about how this fact would affect to the model.

Can be the proposed method applied to detect the mass and the position of the added mass?

Author Response

Round  2

Reviewer 1 Report

The revised manuscript is acceptable for publication.